# Basement membrane proteins as a substrate for efficient *Trypanosoma brucei* differentiation in vitro

**Federico Rojas** *, **Mathieu Cayla**, **Keith R. Matthews**

Institute for Immunology and Infection Research, School of Biological Sciences, University of Edinburgh, Edinburgh, United Kingdom

* federico.rojas@ed.ac.uk

## Abstract

The ability to reproduce the developmental events of trypanosomes that occur in their mammalian host *in vitro* offers significant potential to assist in understanding of the underlying biology of the process. For example, the transition from bloodstream slender to bloodstream stumpy forms is a quorum-sensing response to the parasite-derived peptidase digestion products of environmental proteins. As an abundant physiological substrate *in vivo*, we studied the ability of a basement membrane matrix enriched gel (BME) in the culture medium to support differentiation of pleomorphic *Trypanosoma brucei* to stumpy forms. BME comprises extracellular matrix proteins, which are among the most abundant proteins found in connective tissues in mammals and known substrates of parasite-released peptidases. We previously showed that two of these released peptidases are involved in generating a signal that promotes slender-to-stumpy differentiation. Here, we tested the ability of basement membrane extract to enhance parasite differentiation through its provision of suitable substrates to generate the quorum sensing signal, namely oligopeptides. Our results show that when grown in the presence of BME, *T. brucei* pleomorphic cells arrest at the G0/1 phase of the cell cycle and express the differentiation marker PAD1, the response being restricted to differentiation-competent parasites. Further, the stumpy forms generated in BME medium are able to efficiently proceed onto the next life cycle stage *in vitro*, procyclic forms, when incubated with cis-aconitate, further validating the *in vitro* BME differentiation system. Hence, BME provides a suitable *in vitro* substrate able to accurately recapitulate physiological parasite differentiation without the use of experimental animals.

## Author summary

African trypanosome parasites responsible for human and animal trypanosomiasis, live extracellularly in their mammalian host and exploit environmental information to regulate their virulence and transmissibility. Morphological slender forms that proliferate in the mammalian host transform into the non-proliferative stumpy forms by receiving an extracellular signal generated by the accumulating parasites. Parasites secrete peptidases

**Data Availability Statement:** All relevant data are within the manuscript.

**Funding:** Funding support was received from Wellcome Trust (https://wellcome.ac.uk) grant

103740/Z/14/Z (KM, MC, FR). The funders had no role in study design, data collection and analysis, decision to publish, or preparation of the manuscript.

**Competing interests:** The authors have declared that no competing interests exist.

into the host that degrade extracellular matrix (ECM) proteins. In this study, we evaluated the capacity to generate stumpy forms *in vitro* by including a basement membrane extract into the culture medium. We show that cells exposed to ECM proteins are able to differentiate efficiently into cell-cycle arrested stumpy forms and that the generated differentiation signal depends on the stumpy induction signalling pathway. This method can be used to generate stumpy forms *in vitro*, these being suitable for experimental analysis, thereby reducing the use of mice.

## Introduction

The extracellular matrix (ECM) is a composite of cell-secreted molecules that offers biochemical and structural support to cells, tissues, and organs. ECM contains laminin 1, fibronectin, vitronectin, entactin, heparin sulphate proteoglycan and type IV collagen. Collagen represents the most abundant protein found in mammals and accounts for over 90% of the total protein mass in the ECM [1]. Collagen fibres provide tensile strength to the ECM and maintain scaffolding for cell-to-cell communication. Twenty-eight different collagens containing 46 distinct polypeptide chains have been identified in vertebrates [2]. These different collagens possess a primary structural Gly-Pro-X repeat (with X often being another proline) that is responsible for their characteristic right-handed helix secondary structure [1]. A collagen molecule consists of three subunits called α chains and has a triple-helical structure. This triple helical structure resists most protease digestion except for collagenases such as matrix metalloproteinases (MMPs). Gelatin is a denatured form of collagen and collagen-derived peptides (CP) formed by protease hydrolysis.

Various di- or tripeptides are included in CP. Furthermore, it is thought that collagen in living tissues is degraded into collagen peptides by various enzymes secreted by osteoclasts or osteoblasts during the process of bone metabolism [3]. Several food-derived collagen oligopeptides have also been identified in human blood after oral ingestion of CP, reflecting their serum stability. One contributor to this stability is the presence of Prolyl-hydroxyproline (Pro-Hyp) containing peptides which are relatively resistant to hydrolysis *in vivo* and may play important functions in target tissues [4]. For example, it has been reported that Pro-Hyp affects the growth of fibroblasts and regulates the differentiation of chondrocytes. Additionally, Matrikines, including Pro-Hyp containing peptides, are biomolecules derived from larger ECM macromolecules such as hyaluronan, collagen, or laminin that serve as a physiological signal for cells or cellular receptors that are not activated by the full-sized parent matrix macromolecule [5].

*Trypanosoma brucei spp.*, responsible for human and animal trypanosomiasis, live extracellularly in their mammalian host and exploit environmental information to regulate their virulence and transmissibility. Morphological slender forms that proliferate in the mammalian host transform into the non-proliferative stumpy forms by receiving an extracellular signal generated by the accumulating parasites [6–8]. Molecular components of this quorum sensing (QS) response mechanism have been recently characterised, showing that peptidases released by the parasite into the host are important in generating oligopeptides that trigger differentiation [9]. The identification of oligopeptides as inducers of this transformation allowed us to explore substrates found in the host which these peptidases could be acting upon in order to promote parasite development.

A study of the *T. brucei* secretome has revealed it to be very rich in various peptidases, covering more than 10 peptidase families or subfamilies [10]. Two trypanosome peptidases, oligopeptidase B and prolyl oligopeptidase, have been shown to be insensitive to host serum

inhibitors and retain their catalytic activity in the blood of infected hosts [11, 12]. In *T. cruzi*, prolyl-oligopeptidase has been shown to have collagenase activity against collagens type I and IV, as well as fibronectin [13], suggesting that this enzyme may be important for degrading the extracellular matrix and allowing the parasite to penetrate host tissues, as well for cell invasion [14]. In *T. brucei*, prolyl oligopeptidase hydrolyzes peptide hormones containing Pro or Ala at the P1 position and also cleaves type I collagen *in vitro* and native collagen present in rat mesentery [15]. Metalloproteases also play an important role in the degradation of proteins that constitute the blood-brain barrier, namely collagen, fibronectin, and laminin, all of which are critical components of the vascular matrix. Consequently, Bastos et al. [15] have suggested that the extracellular release of peptidases from *T. brucei* could contribute to the disruption of the blood brain barrier by hydrolysis of collagen and/or peptides resulting from its partial degradation. Released parasite-derived peptidases could also perturb metabolic or endocrine homeostasis through hormonal regulation, since identified substrates of trypanosome peptidases include thyrotropin-releasing hormone (TRH), gonadotropin-releasing hormone (GnRH) and atrial natriuretic factor (ANF) [15, 16].

Taking into consideration that oligopeptides promote slender to stumpy differentiation and that trypanosomes release stable peptidases that can act on extracellular matrix substrates, we explored whether matrix components could support and promote parasite differentiation *in vitro*. Our results demonstrate that a basement membrane extract culture (BME) supplement promotes stumpy formation via the quorum sensing pathway in comparison to other media, including standard liquid broth or methylcellulose containing medium. The use of BME as a culture supplement will facilitate analysis of this developmental step *in vitro* and reduce the requirement for animal use in some experiments.

## Methods

### Trypanosomes

*Trypanosoma brucei* EATRO 1125 AnTat1.1 90:13 (TETR T7POL NEO HYG) [17], *Trypanosoma brucei* AnTat1.1 J1339 [9] or *Trypanosoma brucei* AnTat1.1 J1339 ZC3H20 Knock-out [18] parasites were used throughout for all pleomorphic cell analysis. For monomorphic trypanosome analysis, *T. brucei* Lister 427 90:13 (TETR T7POL NEO HYG) [19] cells were used. Parasites were grown *in vitro* in HMI-9 medium (Life technologies) [20] at 37°C 5% $CO_2$. Antibiotic concentrations used for parasite maintenance were: Hygromycin (0.5μg/ml), puromycin (0.5μg/ml), phleomycin (1.5μg/ml), G418 (2.5μg/ml), Blasticidin (10μg/ml).

### Basement Membrane Extract assays

Cultrex Basement Membrane Extract, PathClear (Trevigen, Gaithersburg, MD) or Geltrex (Thermo Fisher, A1569601) was added to 24-well plates (between 50–300 μl–total protein concentration ~0.6–3.6 mg) and allowed to gelify at 37°C for 30 min. Cells were diluted in 2 ml of HMI-9 and plated on top of the gelled basement membrane extract and cultured for up to 72h. Samples for PAD1 analysis by immunofluorescence were prepared after 72hrs post-inoculation. To allow extraction of cells embedded in the gel, the mixture of HMI-9 and BME was gently passed through a syringe (0.2μm) to disrupt the collagen fibers. Cells were then centrifuged at 4500g for 5 minutes and washed with cold PBS.

### *In vitro* differentiation to procyclic forms

Parasites grown in HMI-9 with or without BME were supplemented with 6mM pH7.0 cis-aconitate (Sigma, A3412) and were incubated at 27°C. Samples were collected for

immunofluorescence analysis at 0h, 3h and 6h post addition of cis-aconitate. Progression to procyclic forms was monitored by their expression of EP procyclin.

## Immunofluorescence and cell cycle analysis

Paraformaldehyde-fixed cells were adhered to Polysine slides (VWR; 631–0107). 20 μl 0.1% triton in phosphate-buffered saline (PBS) was applied to each well for 2 min. This was then aspirated and the wells were washed with a large drop of PBS. The wells were blocked with 2% BSA in PBS for 45 min at 37°C in a humidity chamber. They were then incubated with 20 μl primary antibody (diluted in 2% BSA in PBS, αPAD1 1: 1,000), αEP procyclin 1:500 (VH Bio)) for 1.30 hour at 4°C in a humidity chamber. Positive control wells and secondary antibody-only wells were included for each experiment. The wells were each washed five times by repeatedly applying and aspirating PBS. They were next incubated with 20 μl secondary antibody (diluted in 2% BSA in PBS, α-rabbit (for PAD1) or α-mouse (for EP procyclin) Alexa fluor 488 1:500 (Life technologies)) for 45 min at 4°C in a humidity chamber. 20 μl of a DAPI working dilution (10 μg ml$^{-1}$ in PBS) was then applied to each well for 1 min, followed by five washes with PBS. Slides were mounted with a cover slip by the application of Mowiol containing 2.5% diazabicyclo(2.2.2) octane DABCO and analysed on a Zeiss Axioskop 2 plus or Zeiss Axio Imager Z2. QCapture Suite Plus Software (version 3.1.3.10, https://www.qimaging.com) was used for the image capture. Images of PAD1 and DAPI staining were overlaid in ImageJ 64 and cell counts were performed using the Cell Counter plugin. Cells were considered PAD1 + when expression of this stumpy marker was expressed at the surface of parasites. Counting was not performed blindly. >250 cells were counted per sample and per time point.

## Quantification and statistical

Graphical and statistical analyses were carried out in GraphPad Prism version 9 (GraphPad Software, La Jolla, California, USA, https://www.graphpad.com) by two-way repeated-measures ANOVA test followed by Tukey post-hoc analysis. Graphs provide mean values ± SEM.

## Results

### Evaluation of basement membrane extract containing medium for parasite growth and differentiation

Matrigel (BME) is a soluble and sterile extract of basement membrane proteins derived from the Engelbreth-Holm-Swarm tumour that forms a 3D gel at 37°C and supports mammalian cell morphogenesis, differentiation, and tumour growth [21]. The major components of BME include laminin, collagen IV, entactin, and heparin sulphate proteoglycans.

Pleomorphic *T. brucei* parasites (EATRO 1125, AnTat1.1 90:13)[19] were grown *in vitro* in the presence of BME and the effects on cell proliferation assessed with respect to growth in HMI-9 liquid trypanosome culture medium (Fig 1A). Given the possibility that increasing the viscosity of the medium could induce cells to differentiate, we also tested the ability of cells to differentiate in Geltrex, an extracellular matrix with 10 times less protein concentration than BME but with similar gelling properties, as well as HMI-9 containing agarose 0.65%(w/v) and 1.1%(w/v) methylcellulose, which have been used previously to support pleomorphic cell growth [22, 23]. Cells grew at comparable rates during the first 48 hrs but by 72 hrs parasites in the presence of BME showed a stumpy-like phenotype. BME exposed cells showed accumulation of 98.7%±0.98 of the cells in a 1K1N (1 kinetoplast 1 nucleus), 0.39%±0.07 (2K1N) and 0.9%±0.9 (2K2N) cell-cycle configuration, consistent with the cell cycle arrest observed for stumpy parasites. Meanwhile, cells cultured in HMI-9 showed 89%±3.2 in a 1K1N, 5.5% ±1.2

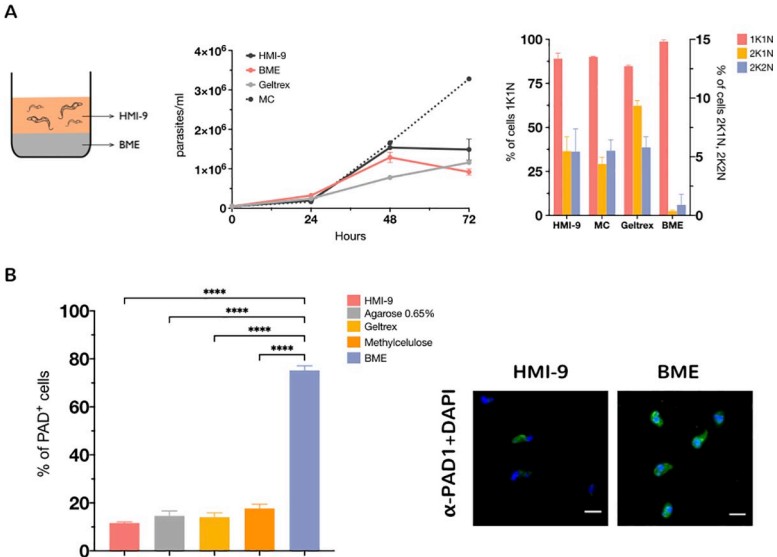

**Fig 1.** (A) Growth of pleomorphic *T. brucei* AnTat1.1 90:13 in the presence of different support media. MC: methylcellulose; BME or HMI-9 as a control. n = 2 per group. The right-hand panel shows the proportion of cells with 1 kinetoplast and 1 nucleus (1K1N, G1 and S phase cells), 2 kinetoplasts and 1 nucleus (2K1N, G2 phase cells) or 2 kinetoplasts and 2 nuclei (2K2N, post mitotic cells) from an analysis of 250 cells per condition at 72hs. (B) Expression of the stumpy marker PAD1 in the presence of different support media. (****$P < 0.0001$, one-way analysis of variance with Tukey's multiple comparisons test). The right-hand panel shows representative images of PAD1 (green) expressing cells at different BME concentrations. >400 cells were counted for each condition. The parasite nucleus and kinetoplast (stained with DAPI) is pseudocoloured in blue. Scale bars, 10 μm.

in a 2K1N and 5.4% ±1.9 in a 2K2N configuration. These results show that including BME in the culture environment promotes cell cycle arrest of *T. brucei* cells *in vitro* (Fig 1, right panel). We then assayed levels of PAD1 expression, a stumpy cell enriched surface marker protein. Parasites grown in the presence of BME showed a significantly higher level of cells expressing PAD1 at 72hrs (75.2% ±1.9, *p<0.0001*), whereas in HMI-9 only 11.6% ± 0.5 of cells exhibited expression of PAD1 at the same time-point. In contrast, culture in the presence of either agarose at 0.65% (w/v) or in 1.1% (w/v) methylcellulose generated much lower levels of PAD1-expressing cells than with BME (14.6% ±2.0 and 17.7% ±1.7, respectively). Moreover, Geltrex did not increase the proportion of differentiated cells (14.0% ±1.8 PAD1+), indicating that the higher concentration of the protein substrate present in BME matrigel is necessary to generate the differentiation signal (Fig 1B).

To analyse the effect of providing different concentrations of BME to the culture media, we tested three different volumes of BME added to the 24-well plates (50, 150 and 300 μl of BME in 2 ml HMI-9) (Fig 2A). Commercial BME has a protein concentration that ranges from 12–18 mg/ml, thus in our experiment parasites were provided with ~0.6–0.9, 1.2–1.8 or 2.4–3.6 mg (in 2ml HMI-9), respectively. Increasing the concentration of BME in the media resulted in a concomitant increase in PAD1 expression (22.5%±3.2 for 50 μl, 63.5%±6.8 for 150 μl, 76.5%±4.6 for 300μl–HMI-9 22.6%±3.5), confirming that matrigel provides the necessary substrates for the generation of stumpy forms in a concentration-dependent manner.

## Stumpy forms generated in basement membrane extract medium differentiate effectively to procyclic forms

We next assessed the capacity of the stumpy forms generated in BME to differentiate to tsetse midgut-like procyclic forms. After 72hrs incubation in BME, cells were exposed to 6 mM *cis-*

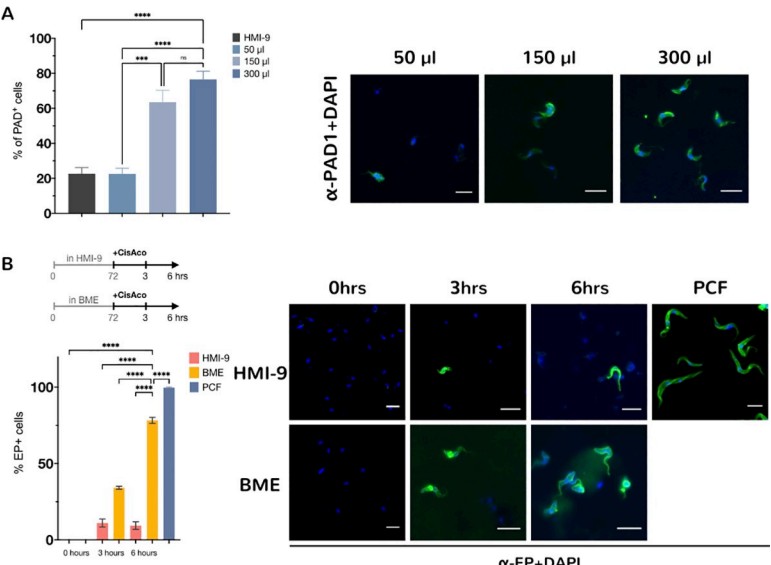

**Fig 2. Expression of the PAD1 marker is dependent on BME concentration.** (A)*T. brucei* AnTat1.1 90:13 cells were grown in the presence of increasing concentrations of BME and samples were taken at 72 hrs. Increasing the concentration of substrates leads to a higher percentage of cells expressing the stumpy marker PAD1. 200 cells counted per condition (\*\*\*\**P* < 0.0001, \*\*\**P* < 0.0002, n.s: not significant; one-way analysis of variance with Tukey's multiple comparisons test). The right-hand panel shows representative immunofluorescence images for cells cultured in the presence of different concentrations of BME with PAD1 pseudo-coloured in green and DAPI (nucleus and kinetoplast) in blue. Scale bars, 10 μm. (B) Expression of EP procyclin after incubation with either HMI-9 alone or BME for 72 hrs and exposure to 6mM cis-aconitate for 0, 3 or 6 hrs from the analysis of >250 cells per condition. The stumpy form parasites obtained in BME express the EP procyclin protein to a higher level than cells isolated from HMI9. Procyclic form cells (PCF) are included as a positive control. Error bars = S.E.M. (\*\*\*\**P* < 0.0001; one-way analysis of variance with Tukey's multiple comparisons test). The right-hand panel shows representative immunofluorescence images of EP+ cells (in green) after exposure to cis-aconitate. The parasite nucleus and kinetoplast (stained with DAPI) is pseudo-colored in blue. Scale bars, 10 μm.

aconitate, a trigger for onward differentiation [24, 25]. 34.1%±0.95 of cells expressed EP procyclin after 3 hr and 78.2% ±1.96 by 6 hr post exposure to *cis*-aconitate (Fig 2B). This was in contrast to the same pleomorphic cells grown in HMI-9 alone and exposed to cis-aconitate (11% ±2.6 at 3 hours and 9.4%±2.5 -EP positive cells at 6 hours post-addition). These results show that the stumpy forms generated by the incubation in the presence of BME are capable of differentiating into the next life-cycle form.

## Basement membrane extract promotes parasite differentiation through the QS- signalling pathway

To confirm that the effect on pleomorphic cells was due to the generation of the differentiation signal, *T. brucei* Lister 427 90:13 monomorphic cells [19], which have lost the capacity for stumpy formation through serial passage *in vitro*, were also incubated in the presence of BME (Fig 3A). Unlike pleomorphic cells, monomorphic cells in BME did not express the stumpy marker PAD1 and grew to similar levels as cells in HMI-9 alone.

The physiological stumpy formation signalling-pathway has been recently characterized in molecular terms [26]. To establish whether the differentiation response was enacted through this pathway in the presence of BME, null mutant cell lines for one of the genes required for quorum sensing signalling were incubated in BME-containing media. TbZC3H20 is a predicted RNA regulator and a proposed substrate for the TbDYRK kinase [18, 27, 28]. Null mutants of TbZC3H20 are insensitive to the QS signal *in vitro* and fail to differentiate

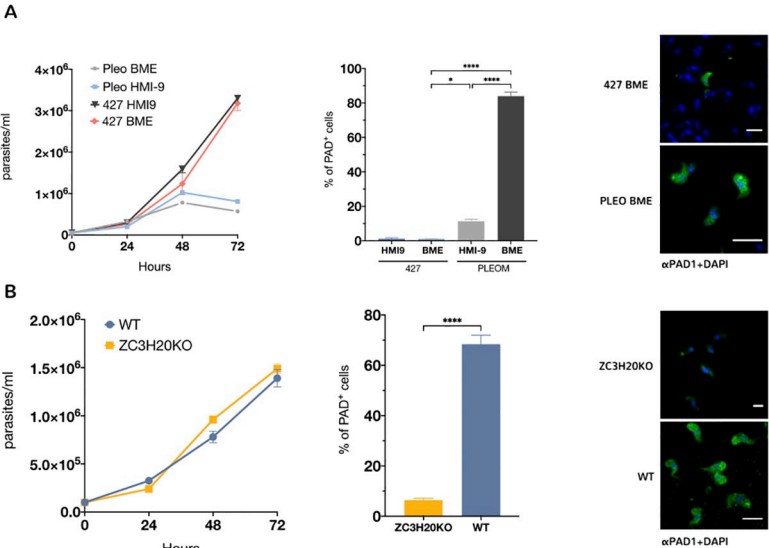

**Fig 3. BME induces differentiation through the SIF pathway.** (A) Growth of monomorphic *T. brucei* 427 vs pleomorphic *T. brucei* AnTat1.1 90:13 in the presence of BME. The middle panel shows the expression of the stumpy marker PAD1 in the presence BME for monomorphic and pleomorphic cells (>500 cells were counted per condition). Error bars = S.E.M. (****$P < 0.0001$; *$P = 0.0169$, one-way analysis of variance with Tukey's multiple comparisons test). The right-hand panel shows representative immunofluorescence images of PAD1+ cells (in green) for monomorphs (427) and pleomorphs (Pleo) cells. Nucleus and kinetoplast (stained with DAPI) are pseudo-colored in blue. Scale bars, 10 μm. (B) Growth of a null mutant line for the stumpy formation signalling pathway component ZC3H20 vs wild type (WT) pleomorphic *T. brucei* AnTat1.1 90:13 in the presence of BME. Unlike WT cells, ZC3H20 KO cells do not express the PAD1 protein as these cells have the stumpy formation signalling pathway disrupted (middle panel). >200 cells counted per condition. ****$P < 0.0001$, one-way analysis of variance with Tukey's multiple comparisons test. The right-hand panel shows representative immunofluorescence images of PAD1+ cells (in green) for *T. brucei* AnTAT 1.1 90–13 cell (WT) and ZC3H20 null mutant (ZC3H20) cells. The parasite nucleus and kinetoplast (stained with DAPI) are pseudo-coloured in blue and kinetoplast (stained with DAPI) are pseudo-coloured in blue. Scale bars, 10 μm.

efficiently into the stumpy forms *in vivo*. When analysed in BME containing medium, the growth of TbZCH20 null mutants was equivalent to the same cells grown in HMI-9 alone. However, PAD1 expression was significantly lower in the null mutant cell line compared to the parental cell line (Fig 3B). This result indicates that the signal generated by incubating cells in BME is acting through the physiological QS pathway, inducing PAD1 expression, cell cycle arrest and parasite differentiation.

## Discussion

Our results show that basement membrane extract can provide suitable substrates for the signal that fuels quorum sensing-mediated cell differentiation in *Trypanosoma brucei*. This is consistent with our previous results where we describe the signal generator (peptidases), the signal (oligopeptides) and the transporter (TbGPR89) involved in the trypanosome's density-sensing mechanism. Importantly, we were able to replicate this developmental process, that normally occurs in the mammalian bloodstream and tissues, in culture conditions by including BME collagen/laminin in the media. Furthermore, the specificity of this substrate in inducing differentiation *in vitro* was demonstrated using developmentally incompetent monomorphic parasites unable to either reduce growth or express a stumpy-specific marker protein, PAD1. The operation of the signal through the physiological quorum sensing signalling pathway was also confirmed by the analysis of a null mutant cell line for the key developmental regulator TbZCH20, a substrate of a known signalling kinase that controls the

developmental programme, TbDYRK. These parasites were unable to differentiate in response of the BME, matching their differentiation defect when analysed *in vivo*. Hence, BME medium induced the known characteristics of stumpy development (growth arrest, accumulation in 1K1N (G0/G1), PAD1 expression) acting through the characterised developmental pathway that functions *in vivo*.

Several short peptide sequences derived from ECM proteins have been used to induce mammalian cell differentiation in culture. For example, the laminin-derived Ile-Lys-Val-Ala-Val (IKVAV) and Tyr-Ile-Gly-Ser-Arg (YIGSR), as well as the fibronectin-derived Arg-Gly-Asp (RGD), are the most commonly used sequences to promote neural cell adhesion, migration and differentiation *in vitro* [29]. Also, collagen-derived dipeptides containing Pro-Hyp (proline-hydroxyproline) promote osteoblastic MC3T3 cell differentiation through upregulation of osteogenic Foxg1 expression [30]. It is possible that in the induction of parasite differentiation, oligopeptides generated by the incubation with collagen/lamin are imported and bind receptor proteins inside the cytosol of *T. brucei*, inducing a cascade of events leading to cell cycle arrest and metabolic changes. This awaits exploration.

Some peptides are frequently degraded within a short time by serum peptidases *in vivo*. Hydroxyproline-containing peptides are far more stable in the body, given that enzymatic systems able to hydrolyse the corresponding peptide bonds occur rarely [31]. The stability of these oligopeptides in blood makes it plausible for them to act as a differentiation signal and reach systematically beyond the vicinity where the parasite peptidases are released. Alternatively, other protein degradation products could provide the physiological signal locally or systemically and, indeed, we have already demonstrated that there is specificity to the oligopeptide signals that promote differentiation such that some may be able to signal more effectively than others [9]. Hence, the combination of sensitivity to particular oligopeptides and their specific stability may contribute to a complex signal threshold dependent upon the precise combination of different oligopeptides and their local concentration and turnover.

Previous studies show that trypanosomatids have a close interaction with collagen and other proteins that form part of the ECM [32], bringing the secreted peptidases and their substrates into close proximity. Interestingly, it has been shown that *T. cruzi* trypomastigotes exhibit extensive proteome remodelling when incubated with ECM (Geltrex)[33]. In that case, ECM triggers a reduction in the glycolytic enzyme activity (hexokinase/glucokinase and pyruvate kinase) and a reprogramming of the trypomastigote metabolism. Similarly, when *T. brucei* differentiates into stumpy forms, a shift of the metabolism toward acetate formation occurs [34]. Thus, in a mammalian infection, given the abundance of collagen/laminin in tissues, it is likely that peptidases released by infecting *T. brucei* can generate dipeptides/tripeptide signals that act as a proxy for the increasing parasitaemia in the host. This mechanism would limit the increase in number of replicating parasites and so prevent killing of the infected host.

To summarise, we have extended and corroborated our previous results that oligopeptides promote QS signalling, and demonstrated that these oligopeptides can originate from ECM proteins. In addition, we demonstrate that ECM extract can be used to generate stumpy forms *in vitro* via the physiological QS sensing pathway, with these being suitable for experimental analysis. For example, this approach could be used to rapidly evaluate pleomorphic RNA interference or null mutant cell lines for developmental defects before they are validated using an *in vivo* model. Growth in BME medium can also be used to study the efficient differentiation process from stumpy forms to procyclic forms without the need to obtain stumpy forms from mice, contributing to methods that enable complete recapitulation of the trypanosome life cycle *in vitro* and reducing animal usage in functional studies of trypanosome development. The use of a complex yet consistent extracellular matrix substrate will also facilitate the characterisation of the peptidase and oligopeptide signals that promote differentiation in African

trypanosomes. This provides a more defined and reproducible model for proteomic or metabolomic studies where parasites are maintained in a more physiological condition than HMI-9 liquid broth or other media.

## Author Contributions

**Conceptualization:** Federico Rojas, Keith R. Matthews.

**Data curation:** Federico Rojas, Keith R. Matthews.

**Formal analysis:** Federico Rojas, Keith R. Matthews.

**Funding acquisition:** Keith R. Matthews.

**Investigation:** Federico Rojas, Mathieu Cayla.

**Methodology:** Federico Rojas, Mathieu Cayla.

**Supervision:** Keith R. Matthews.

**Visualization:** Federico Rojas.

**Writing – original draft:** Federico Rojas, Keith R. Matthews.

**Writing – review & editing:** Federico Rojas, Mathieu Cayla, Keith R. Matthews.

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
