## [Decision Letter · Decision Letter 0]

22 Mar 2021

Dear Dr Rojas,

Thank you very much for submitting your manuscript "Basement membrane proteins as a substrate for efficient Trypanosoma brucei differentiation in vitro" for consideration at PLOS Neglected Tropical Diseases. As with all papers reviewed by the journal, your manuscript was reviewed by members of the editorial board and by several independent reviewers. The reviewers appreciated the attention to an important topic. Based on the reviews, we are likely to accept this manuscript for publication, providing that you modify the manuscript according to the review recommendations. 

Both Reviewers have raised significant questions and need to be responded.

Sincerely,

Hira L Nakhasi

Associate Editor

Paula MacGregor

Deputy Editor

Both Reviewers have raised significant questions and need to be responded.

Reviewer's Responses to Questions

**Key Review Criteria Required for Acceptance?**

**Methods**

-Are the objectives of the study clearly articulated with a clear testable hypothesis stated?

-Is the study design appropriate to address the stated objectives?

-Is the population clearly described and appropriate for the hypothesis being tested?

-Is the sample size sufficient to ensure adequate power to address the hypothesis being tested?

-Were correct statistical analysis used to support conclusions?

-Are there concerns about ethical or regulatory requirements being met?

Reviewer #1: In this manuscript the authors have set out to develop a robust in vitro system to support Trypanosoma brucei differentiation from slender to stumpy and then onto procyclic forms. The methodology of this paper is appropriate to address the objectives of this study. Moreover, this will be an incredible useful approach for the dissection of this differentiation process.

Reviewer #2: The study is well carried out and the statistically analyses and N numbers clearly explained. 

The PAD1 and EP pos/neg counting is done by microscopy scoring which is susceptible to bias. The authors should indicate whether the counting was done blinded or give more details on their criteria regarding thresholding etc.

**Results**

-Does the analysis presented match the analysis plan?

-Are the results clearly and completely presented?

-Are the figures (Tables, Images) of sufficient quality for clarity?

Reviewer #1: The results are are clearly presented in the figures.

Reviewer #2: The authors use tried and tested methods to determine the effect of BME upon slender-stumpy differentiation in culture. The results are unambiguous, well presented and support their conclusions. The dose response experiments (Figure 2A) are especially compelling. 

The authors should extend this to determine whether increases in stumpy form differentiation efficiency levels out above a threshold BME concentration.

Can the authors comment on why the pleomorphic cells have similarly retarded growth in BME and methylcellulose and geltrex, but BME supports far higher PAD1 pos cells and cell cycle arrest? (Figure 1)

Figures:

Figure 1: growth curve should be log scale (and others throughout the manuscript)

Figure 2: a 0hrs timepoint should be included

**Conclusions**

-Are the conclusions supported by the data presented?

-Are the limitations of analysis clearly described?

-Do the authors discuss how these data can be helpful to advance our understanding of the topic under study?

-Is public health relevance addressed?

Reviewer #1: Overall, the data in the paper supports the conclusion and the graphs and images are clear and well presented. The authors are able to show that the addition of BME to the media supports differentiation of pleomorphic trypanosomes through the quorum-sensing pathway.

Reviewer #2: The authors’ conclusions are supported by the data – i.e. BME supports development of stumpy forms better than other media additives and it is likely to be a good method for studying differentiation in trypanosomes.

A discussion on whether this system is likely to allow in vitro investigation into quorum sensing in related African trypanosomes would be worthwhile.

**Editorial and Data Presentation Modifications?**

Reviewer #1: I only have a few minor comments -

1) Can the authors provide the number of cells counted and the replicates performed for each experiment in the figure legend – some of this information is in the methods but not all and I find it simpler if this is included with the figures.

2) How were the growth analyses performed – can the authors add detail. Are these cumulative growth curves? Were the cells split each day?

3) The balance of the paper seems odd there are only 5 pages of results which includes the figure legends yet there is also 5 pages of discussion – this seems excessive. In the discussion there is lots of detail about collagen (as there was in the introduction) and various peptides that can form and their half-lives etc. I feel this though interesting would be more appropriate for a review of this topic rather than as a discussion of this paper, which is essentially a technology development paper. I would recommend that the authors reduce the length of the discussion to give it greater focus.

Reviewer #2: (No Response)

**Summary and General Comments**

Reviewer #1: (No Response)

Reviewer #2: This study demonstrates that BME can act as a media supplement that can support the efficient differentiation from slender to stumpy forms in pleomorphic Trypanosoma brucei. The authors use various assays (cell cycle arrest, stumpy molecular markers and lifecycle progression) to show this successfully.

This is a useful methods paper that will facilitate / promote the study of slender-stumpy form differentiation by providing a superior in vitro culture system. The advantages of such as system are two-fold: 1) This system has the potential to be far more tractable than in vivo infection. 2) this will reduce the number of mice that must be sacrificed.

This will be of interest to those studying trypanosome differentiation.

PLOS authors have the option to publish the peer review history of their article (what does this mean?). If published, this will include your full peer review and any attached files.

Reviewer #1: No

Reviewer #2: No

Figure Files:

Data Requirements:

Reproducibility:

References

---

## [Editor Report · Decision Letter 1]

16 Apr 2021

Dear Dr Rojas,

We are pleased to inform you that your manuscript 'Basement membrane proteins as a substrate for efficient Trypanosoma brucei differentiation in vitro' has been provisionally accepted for publication in PLOS Neglected Tropical Diseases.

Best regards,

Hira L Nakhasi

Associate Editor

Paula MacGregor

Deputy Editor

---

## [Editor Report · Acceptance letter]

23 Apr 2021

Dear Dr Rojas,

We are delighted to inform you that your manuscript, "Basement membrane proteins as a substrate for efficient Trypanosoma brucei differentiation in vitro," has been formally accepted for publication in PLOS Neglected Tropical Diseases.

Best regards,

Shaden Kamhawi

co-Editor-in-Chief

Paul Brindley

co-Editor-in-Chief
